# Current Status of the Open-Circuit Voltage of Kesterite CZTS Absorber Layers for Photovoltaic Applications—Part I, a Review

**DOI:** 10.3390/ma15238427

**Published:** 2022-11-26

**Authors:** Iulian Boerasu, Bogdan Stefan Vasile

**Affiliations:** 1Lasers Department, National Institute for Lasers, Plasma and Radiation Physics, Atomistilor 409, 077125 Magurele, Romania; 2National Research Center for Micro and Nanomaterials, University Politehnica from Bucharest, 060042 Bucharest, Romania

**Keywords:** CZTS, solar cell, open circuit voltage

## Abstract

Herein, based on the reviewed literature, the current marketability challenges faced by kesterite CZTS based-solar cells is addressed. A knowledge update about the attempts to reduce the open circuit voltage deficit of kesterite CZTS solar cells will be addressed, with a focus on the impact of Cu/Zn order/disorder and of Se doping. This review also presents the strengths and weaknesses of the most commercially attractive synthesis methods for synthesizing thin kesterite CZTS films for photovoltaic applications.

## 1. Introduction

Nowadays, solar cells based on Cu(In, Ga)(S, Se)_2_ (CIGS) and CdTe thin-film solar cells have already reach the commercial stage, having an efficiency of 23.4% for CIGS and 21.0% for CdTe. However, their marketability has stagnated, not only because of the rapid reduction in the price of Si-based photovoltaics but also because these devices include cadmium, which is a toxic element. Moreover, In and Ga are also in the category of critical raw materials (CRM) for the industry, so they are subject to substantial competition for mass production, thus restricting this material for mass production of giga- and tera-Watt power. Some of the others chemical elements forming the CIGS structure, i.e., tellurium and indium, are relatively scarce and, consequently, highly expensive.

A promising solution for a non-toxic and commercially attractive absorber for photovoltaic applications is offered by the family of kesterite semiconductor materials such as copper–zinc–tin–sulfide (with the chemical formula Cu_2_ZnSnS_4_) (CZTS) and copper–zinc–tin–selenide (with the chemical formula Cu_2_ZnSnSe_4_)(CZTSe) and their alloy family copper–zinc–tin–sulfo–selenide (Cu_2_ZnSn(S_x_,Se_1−x_)_4_ (CZTSSe), where 0 ≤ x ≤ 1). The kesterite constituent elements are less toxic and more commonplace elements: copper (Cu), sulfur (S), and zinc (Zn) are 26th, 17rd, and 25th most abundant elements in the Earth’s crust (abundance of elements in the Earth’s crust, https://www.coursehero.com/study-guides/geology/reading-abundance-of-elements-in-earths-crust/ (accessed on 1 November 2022). However, kesterite CZTS solar cells have received less attention than their CIGS counterparts, as their best efficiency has plateaued at around 13% [1]. Shafi et al. used the software package SCAPS-1D to simulate the impact of the ZnO buffer layer processing conditions on “back contact/CZTS/CdS/ZnO/fFront contact/glass” models of solar cell under ideal conditions in terms of temperature, humidity, and dust [2]. The author reported an increase in the ZnO deposition cycles from 2 to 10 turns, an efficiency improvement of 8.503% to 13.222%. Accordingly, notable simulated improvements in performance were reported by Shafi regarding the VOC (i.e., from 0.866 V to 0.887 V), the short circuit current (JSC) (i.e., from 14.860 mA/cm^2^ up to 21.074 mA/cm^2^), and for the fill factor (FF) (i.e., from 66.035% up to 67.491%). Moreover, the certified world record power conversion efficiency of 12.6% was achieved by a CZTS solar cell [3], but this is still far lower than the theoretical predicted by the Shockley–Queisser maximum CZTS power conversion efficiency of 32.2% [4]. It has been proposed, and partially accepted by the kesterite scientific community, that among the device parameters diminishing the efficiency of CZTS, the main factor is the open circuit voltage deficit (ΔV_OC_). This parameter is defined as the difference between the theoretical open-circuit voltage at the Shockley−Queisser limit (V_OC-SQ_) and the measured device’s open-circuit voltage (V_OC-CZTS_) [5]:ΔV_OC_ = V_OC-SQ_ − V_OC-CZTS_(1)

In this perspective, the major scientific effort has been directed towards reducing the open-circuit voltage deficit (ΔVoc) to below 345 mV, which characterizes the current world record efficiency of 12% for CZTS solar cells [3,6]. According to Equation (1) [7], lowering the ΔV_OC_ can be achieved only through increasing the CZTS absorber’s V_OC-CZTS_. Consequently, comprehensive studies on the fundamental understanding of existing the physical limitations of V_OC-CZTS_ have been reported in the reviewed literature. These physical issues limiting voltage include bulk point defects and clusters of defects, electron–hole (e-h) non-radiative recombination at grain the boundaries, and the formation of voltage-robbing secondary phases in the bulk absorber [7,8,9,10,11,12]. However, as the ΔV_OC_ of the final photovoltaic device based on kesterite CZTS absorbers layers could be restricted by many others different factors, such as non-ideal p-n heterojunction formation, band offsets between CZTS(Se) and the buffer layer, and back contact issues [8,13,14], it is difficult to pinpoint the bulk absorber’s detrimental characteristics as the universal origin of the ΔV_OC_’s failures.

In this review, we summarize some of the most notable information collected in the literature about V_OC-CZTS_ with respect to the bulk CZTS absorber’s composition and some of the most mature processing technologies.

## 2. Impact of Bulk Properties on Open-Circuit Voltage

Kesterites belong to the $A_{2}^{I}B^{\mathrm{II}}C^{\mathrm{IV}}X_{4}^{\mathrm{IV}}$ family, with a tetragonal structure [15]. The structure of kesterite consists of alternating layers of Cu–Sn, Cu–Zn, Cu–Sn, and Cu–Zn at z = 0, ¼, ½, and ¾, respectively. Accordingly, the Cu atom occupies the 2a (0, 0, 0) position while Zn and other Cu atoms occupy 2c (0, ½, ¼) and 2d (0, ½, ¾), resulting in the kesterite CZTS phase with the space group $\bar{I4}$. In the chemical equilibrium of CZTS, the forward and backward reactions are produced at the same rate at the atomic and molecular levels, with no net change in the concentration of the products and reactants [16]:Cu_2_S(s) + ZnS(s) + SnS(s) + 1/2S_2_ <=> CZTS(s)(2)
SnS(s) <=> SnS(g)(3)

However, Cu and Zn cations have almost identical ionic radii (145 pm and 142 pm, respectively) and they both have a tetrahedral coordination. Moreover, a large size and chemical mismatch exists among the cations of Sn, Cu, and Zn. Consequently, Cu and Zn atoms can swap places at a fairly low enthalpy cost, leading to the formation of a high density of anti-site defects (e.g., Cu_Zn_, Zn_Cu_) and the related thermodynamically stable defect clusters. As those defects primarily involve the Cu and the Zn atoms, this type of disorder is also called “Cu/Zn disorder” within the space group $\bar{I4}m$ [17]. In this $\bar{I4}m$ disordered kesterite, the Cu atoms fully occupy the 2a Wyckoff position (the z = 0 plane), and the Cu an Zn atoms are mixed only in the 2c and 2d Wyckoff positions (the z = 1/4 and 3/4 planes) [15,18,19,20]. CZTS is intrinsically a p-type semiconductor due to the presence of the acceptor defects (i.e., copper vacancies (V_Cu_) and copper-on-zinc (Cu_Zn_) anti-site defects characterized by lower formation energies than the donor defects [21]. Washio et al. reported on the substitution of Cu by Zn at the 2a site as the Cu/(Zn + Sn) ratio decreases [22]. Cu vacancies have also been reported on the 2a site by Chobrac et al. [23]. Several authors have reported that while shallow defects, e.g., V_Cu_, are beneficial to enhancing the p-type conductivity, the deep-level defects, e.g., Cu_Zn_, can effectively lower the bandgap of the CZTS material and, consequently, lower the V_OC-CZTS_ [23,24,25]. Moreover, the recombination rate increases linearly with the concentration of deep-level defects. It has been reported that that the main recombination process in a CZTS material is band to impurity rather than band to band. Gokmen [5] reported that band tailing occurs in CZTS as a consequence of the electrostatic potential fluctuations associated with the formation of defects in the complexes, such as [${\mathrm{Cu}}_{\mathrm{Zn}}^{-}$ + ${\mathrm{Zn}}_{\mathrm{Cu}}^{+}$]. Another potential fluctuation issue is related to the existence of secondary phases. CZTS is a complex material, with many components. This being so, its phase stability region is narrow and, depending on the deposition conditions, binary and ternary phases, such as ZnS, Cu_x_S, SnS_x_, or Cu_2_SnS_3_, can co-exist with the quaternary Cu_2_ZnSnS_4_ phase of kesterite. The coexistence of mixed phases can drastically contribute to the mitigation of the devices’ performance. 

The main challenge in processing CZTS absorber layers is related to the issue of synthetizing the single phase of kesterite because of the very narrow window of chemical potential that is favorable for the formation of the quaternary phase of Cu_2_ZnSnS_4_ [26,27]. As seen in Figure 1, the stable region of chemical potential which is favorable is 1 eV long and 0.1 eV wide. 

Moreover, as shown in Figure 1, the stable region of the kesterite phase is shifted to the ZnS–SnS_2_ boundary of the phase diagram. These CZTS compositions are usually called “Cu-poor”, e.g., Cu/(Zn + Sn) = 0.74 ÷ 0.8, and “Zn-rich”, e.g., Zn/Sn = 1.2 ÷ 1.36 [27,28,29,30]. Lafond et al. reported the results of his study on the compositional ability of the kesterite CZTS and its flexibility to deviate from stoichiometry (Cu:Zn:Sn = 2:1:1) [15,23,31]. Based on the charge balance, Lafond used a model of four cation substitution reactions and proposed four off-stoichiometry types that are formed by incorporation of defect complexes in a kesterite CZTS structure: (i) A-type Cu-poor/Zn-rich, where the Cu is substituted, forming a Cu vacancy (V_Cu_) and Zn on the Co anti-site (Zn_Cu_); (ii) B-type Cu-poor/Zn-rich where Cu and Sn are substituted by Zn forming Zn on Cu (Zn_Cu_) and Zn on Sn (ZnSn); (iii) C-type Cu-rich/Zn-poor, in which Zn substitutions form Cu on Zn (Cu_Zn_) and Sn on Zn (Sn_Zn_) defects; and (iv) D-type Cu-rich/Zn-poor, where Cu substitutes for the Zn forming Cu on the Zn anti-site (Cu_Zn_) and additional interstitial Cu (Cu_i_). Lafond’s classification has lately been extended by Dimitrievska et al. [32]. Dimitrievska has also reported on the impact of Cu-substitution defects on the photovoltaic properties of CZTS, especially V_OC_. The authors concluded that the values of V_OC_ could be tuned by clusters of Cu substitution defects, particularly ⎣$V_{\mathrm{Cu}}+{\mathrm{Zn}}_{\mathrm{Cu}}$⎦, either by tuning the composition or altering the process of synthesis.

Further progress was achieved by Valle Rios et al. [15]. Assuming a Sn^4+^ → Zn^2+^ + 2Cu^2+^ cation substitution, the authors reported the identification of either a ZnSn + 2Cu_i_ or CuSn + Cu_i_ + Zn_i_ defect complex, including interstitial Cu_i_ and Zn_i_, in an off-stoichiometry Cu-rich and Zn-rich composition. Figure 2 plots the literature data about the identified secondary phases (Cu_2_Se, ZnSe, and SnSe_2_) and the associated charge-compensated point defect clusters (A, B, C, D, E, and F). 

In the critical review released by Giraldo et al. [10], the authors addressed the questions about why in the results reported by Dimitrievska [10], here plotted in Figure 3, the highest efficiencies were obtained for Cu-poor and Zn-rich CZTS compositions, while their performances fell very close to that of the Cu-rich, Zn-poor, and Sn-stoichiometric composition (“A-line”).

The authors concluded that in CZTS photovoltaic devices, deep defects, inhomogeneity, band tailing, and interface recombination are of the greatest relevance for interpreting the limitations in V_OC_. The formation with the highest probability has V_Cu_, Cu_Zn_, and Zn_Cu_ point defects. These defects are shallow defects (V_Cu_ and Cu_Zn_ are shallow acceptors and Zn_Cu_ is a shallow donor); consequently, these have a limited impact on recombination processes. A commonly adopted strategy for improving the V_OC-CZTS_ consists of promoting the formation of the beneficial shallow V_Cu_ defects while inhibiting the anti-site defects, (i.e., Cu_Zn_ defects) and Cu-rich secondary phases. This strategy could be achieved by intrinsic alkali doping of the CZTS compounds placed in the Cu-poor and Zn-rich region of the Cu–Zn–Sn phase diagram rather than the stoichiometric composition.

Although considerable research efforts have been made, the impact of Cu/Zn order/disorder on the V_OC_ is still a matter of debate. Answering the question whether or not the Cu/Zn order/disorder is responsible for the high experimental V_OC_ deficit could represent a step towards the next major breakthrough of kesterite solar cells. Using theoretical calculations, He et al. demonstrated that the anti-sites defect clusters of Cu_Zn_ + Zn_Cu_ are characterized by having the lowest value of energy formation, indicating that a large population density of these disordering clusters is present in the kesterite lattice, leading to potential and/or bandgap fluctuations and to a large V_OC_ deficit [7,34]. In contrast, using two different models applied within the Shockley–Queisser detailed balance approach, Bourdais et al. [17] concluded that the effect of disorder in the Cu/Zn sublattice seems to have no significant effect upon the V_OC_ of kesterite CZTS devices. It has been proposed that the large experimental deficit in V_OC_ in kesterite solar cells has to be attributed to the strong band tailing originating from large densities of variably compensated point defects [17]. Romero et al. [28] published the results of their study about the relationship between the luminescence and intrinsic point defects in the kesterite Cu_2_ZnSnS_4_. According to these results, a decrease in the intensity of the luminescence, in tandem with the high excitation densities needed to transition from band–tiling recombination to band–bending recombination and then meeting the high injection conditions, could be attributed to the formation of point defects and the related electronic states near the mid-gap, resulting in more a nonradiative recombination. Moreover, the observed redshift in the luminescence from the grain boundaries in CZTS were associated, to some extent, to the formation of a high neutral barrier and low recombination.

According to the theoretical model published by Shockley and Queisser [4], the absorber energy gap *E_g_* represents the upper limit of a cell’s open-circuit voltage *V_OC_*
(4)$$V_{OC}=\frac{E_{g}}{q}-\frac{nk_{B}T}{q}\ln\left(\frac{J_{SC}}{J_{00}}\right)$$
where *E_g_* is the bandgap, *n* is the ideality factor, *k_B_* is the Boltzmann constant, *T* is the temperature, *q* is the elementary charge, *J_SC_* is the pre-exponential factor of the dark saturation current density, and *J_sc_* is the short circuit current density under AM1.5G (a simulated terrestrial solar spectrum). According to the results of hybrid functional calculations [35] mixed with Hartree–Fock exchange or according to the many-body perturbation theory [35], in the case of Cu_2_ZnSnS_4_, the estimated Eg is of 1.5 eV, which is 0.5 eV higher than the value 1.0 eV calculated for Cu_2_ZnSnSe_4_. Persson et al. [36] analyzed the electronic structure as well as the optical response of Cu_2_ZnSnS_4_ and Cu_2_ZnSnSe_4_. Using a relativistic full-potential linearized augmented plane wave method, Persson et al. reported that the fundamental bandgap energy was estimated to be 1.5 eV for Cu_2_ZnSnS_4_ and 1.0 eV for Cu_2_ZnSnSe_4_. 

A survey of the literature on the most efficient kesterite solar cells reported to date revealed that the experimentally bandgap (E_g_) of pure sulfide CZTS varies from 1.53 to 1.67 eV [36,37,38,39,40,41,42], while a domain variation from 0.96 to 1.1eV has been reported for pure selenide absorbers [43,44,45,46,47]. A possible strategy to mitigate the issue of lower V_OC_ in CZTS absorbers is to tune the bandgap of the parent kesterite CZTS by partial substitution of S with Se, leading to the formation the a pentanary alloy system Cu_2_ZnSn(S_1−X_Se_X_)_4_ (CZTSSe), also called sulfoselenide, where 0 ≤ X ≤ 1 [35,48]. The foundation of this approach is based on the compositional nature of the kesterite CZTS family of materials. As the quaternary Cu_2_ZnSnS_4_ (CZTS) selenide derives from the quaternary Cu_2_ZnGeS_4_ compound in which the Ge^4+^ cation replaces the isovalent Sn^4+^. Moreover, if S^2−^ is replaced by isovalent Se^2−^, then the quaternal compound Cu_2_ZnSnSe_4_ (CZTSe) selenide is obtained. Using an ab initio calculation under the assumption of a homogeneous distribution of anions within the absorber layer, Chen et al. modeled the optical bandgap of a CZTS_x_Se_1−x_ alloy system in complete thermodynamic equilibrium by partially replacing the S with Se, as per Equation (5) [17]
(5)$$E_{g_{CZTS_{x}Se_{1-x}}}=xE_{g_{CZTS}}+\left(1-x\right)E_{g_{CZTSe}}-bx\left(1-x\right)$$
where $E_{g_{\mathrm{CZTS}}}$ is the bandgap of pure sulfide CZTS, $E_{g_{CZTSe}}$ is the bandgap of pure selenide CZTSe, and *b* is the bowing coefficient, which is independent of the composition and is equal to 0.01. Kumar et al. reported the results of a theoretical study on quantizing the impact of Se substitution on the *E_g_* in the absorber family of Cu_2_ZnSn(S_1−X_Se_X_)_4_ [8]. Figure 4 summarizes Kumar et al.’s calculated values of *E_g_* and *V_OC_* for solar cells fabricated with a different Se substitution content in the Cu_2_ZnSn(S_1−X_Se_X_)_4_ absorber layer.

As shown in Figure 4, the *V_OC_* increases with the bandgap under a certain amount of Se doping, after which, a drop in *V_OC_* will appear, while the *E_g_* is monotonically increasing with increasein the Se content. 

## 3. Processing Technologies for Thin Kesterite CZTS Absorber Films

Different methods are reported in the literature for the synthesis of thin CZTS absorber films. The most commonly used methods for the synthesis of kesterite CZTS absorbers consist of the growth of thin films via physical or chemical methods.

The physical based methods offer several advantages for processing thin CZTS films, such as control over the deposition rate, high crystallinity, and control over the structural and morphological properties of the film. However, the requirements for high-vacuum technology, the high equipment cost, and the complicated systems involved are the main disadvantages of the vacuum techniques in the industrial mass production of CZTS solar cells.

The chemical methods of synthesizing CZTS absorbers have several features making them very attractive for large-scale industrialization. Between the two types of method, the chemical method offers simplicity of use, low capital investment, compact equipment, and low wastage of raw materials, which are the most notable advantages. However, the chemical methods has some critical issues that restrict the mass production of CZTS solar cells. Among these, a multiple uniform coating is needed to achieve the desired thickness of the CZTS absorber. Consequently, several issues related to the layers’ uniformity and cracks have to be overcome when the sol-gel and electrodeposition methods are involved. Regarding the hydrothermal and solvothermal methods, the main disadvantages are related to the time costs, as well as the high-pressure and high-temperature processing conditions, the lack of control over particle size, and the monodispersity of the products. It is worth pointing out that a record efficiency of 12.6% was reported for the CZTS absorber synthetized by the chemical solution approach using hydrazine as a solvent [3]. However, due to the fact that hydrazine is a hazardous, unstable, and reactive solvent, the large-scale fabrication of CZTS solar cells is critically restricted. 

### 3.1. Physically Based Methods of Synthesizing CZTS Absorbers 

Vacuum deposition methods are physical deposition methods that allow technicians to grow thin films atom-by-atom or molecule-by-molecule on a solid substrate. The key elements of a growing process through a physical method include a high-vacuum deposition chamber which holds the bulk parent raw material, also called the target, from which the constituent atoms or molecules are released as a vapor stream, and the deposition substrate on which the vapor stream collides, leading to the growth of a layer with the same stoichiometric composition as the parent material. Several vacuum-based approaches of synthesizing kesterite CZTS absorbers have been reported in the literature, such as thermal evaporation, sputtering, and pulsed layer deposition [49].

#### 3.1.1. Thermal Evaporation

In the case of a thermal evaporation method, the deposition process consists of heating up the parent raw material to a temperature above its boiling point while keeping the substrate at a substantially lower temperature. Figure 5 shows the basic system used for the thermal deposition of thin films. 

The heated target releases atoms or molecules, which accumulate near the parent surface, forming a vapor cloud. Because of the large temperature difference between the target and the substrate, a temperature gradient occurs, leading to a stream of the vapor from the target to the substrate’s surface. Once the atoms or molecules hit the substrate’s surface, they will condense on it back to a solid state. After applying this deposition method, in 2010, Wang et. al. from IBM reported 6.8% efficiency in Cu_2_ZnSnS_4_ solar cells on glass substrates made by thermal evaporation of Cu, Zn, Sn, and S [30]. Later on, in 2011, Shin et al. from IBM reported a solar cell efficiency of 8.4% for a solar sell based on 600-nm-thick CZTS layers of Cu, Zn, Sn and S deposited by thermal co-evaporation at 150 °C, followed by 5 min high-temperature annealing (570 °C) of the deposited layer under atmospheric pressure [50]. It is worth noting that Shin et al.’s reported efficiency values were independently confirmed and certified by an external, accredited laboratory, i.e., The Newport Technology and Applications Center’s Photovoltaic Laboratory, as having the worldwide record efficiency for pure sulfide CZTS absorbers using any technique at that time. 

Through use of a thermal co-evaporation method, a certified power conversion efficiency of 11.6% for a pure Cu_2_ZnSnSe_4_ solar cell and a significant improvement in the open-circuit deficit of 0.578 V was reported by Lee et al. [51].

#### 3.1.2. Sputtering Methods

Sputtering is a physical deposition method used to growth thin films on a substrate in a vacuum chamber filled with a chemically inert gas, usually argon. Figure 6 shows the conceptual layout of the DC sputtering system usually involved in the deposition of CZTS films.

The sputtering system consists of a high-vacuum chamber, inside which is a pair of electrodes. The top electrode is covered with the material to be transferred, also called the target, while the bottom electrode represents the substrate holder. 

In between the target and substrate, a voltage source is applied, with the negative polarity on the target. Thus, the target becomes a cathode liberating electrons. These electrons collide with the outer electrons surrounding the gas atom, and thus, a hot gas-like plasma phase consisting of ions and electrons is promoted. Once the electrons have been lost, the neutral atom turns into a high-energy ion. The negatively charged source material attracts these positive ions, which fly at it with such high velocity that atomic-sized particles are “sputtered off,” or detached. The positively charged ions thus formed are accelerated to the target and strike with sufficient kinetic energy to dislodge the atoms or molecules of the target material. The liberated atoms or molecules will fly to the substrate’s surface. When a large number of atoms collide on the substrate, they start to form a bond with each other at the molecular level, leading to the growth of a thin film with the same composition as the target material.

The first use of the sputtering method to fabricate CZTS absorbers was reported by Ito et al. in 1988. The authors sputtered a CZTS film from the target material by applying the method of atomic beam sputtering and obtained a CZTS film with a 1.45 eV optical bandgap and an V_OC_ of 165 mV under AM1.5 illumination, without a post-deposition sulfurization treatment and even at a very low substrate temperature of 90 °C [37]. Later on, Ito et al. reported an improvement in the open-circuit voltage to 265 mV by annealing the device in air [52]. In 2007, a 5.74% conversion efficiency was reported by Jimbo et al. for a solar cell based on 480 nm high-quality CZTS films fabricated at a Cu/(Zn + Sn) ratio of 0.87 in a co-sputtering system with three RF sources followed by vapor phase sulfurization [53]. The record laboratory-reported efficiency was reported by Yan et al. [54]. The authors deposited sulfide kesterite CZTS precursors by co-sputtering Cu/ZnS/SnS material using a magnetron sputtering system. The final CZTS films were synthesized by sulfurization of the precursors within a combined sulfur and a SnS atmosphere by rapid thermal annealing at 560 °C for three minutes. The authors demonstrated efficient sulfide kesterite CZTS solar cells with 11% for cells with a small area (0.23 cm^2^) and 10% for a standard-sized cell (1.11 cm^2^). 

#### 3.1.3. Pulsed Laser Deposition

Pulsed laser deposition (PLD) is a simple and versatile vacuum-based technique for depositing thin films of a wide range of materials on a wide variety of substrates at temperatures starting from room temperature. The PLD technique offers some unique features such a high deposition rate, relatively easier transfer of species from the target to the substrate, growth from an energetic beam, and reactive deposition. The PLD technique is used to grow engineered layered materials with a controlled thickness and metastable phases [55].

As shown in Figure 7, the PLD system consists of a high-vacuum chamber in which a high-powered laser beam of a certain energy density is focused on a rotating target. The extreme energy (1 ÷ 5 J/cm^2^) of the focused beam is absorbed by the target’s surface area, leading to the breakdown of chemical bonds within the target material. Consequently, ions, electrons, atoms, radicals, or clusters are ejected from the target’s surface, leading to the formation of a so-called ablation plume. These released species traveling at high speed (around 106 cm/s) through the vacuum chamber until they collide at high impact energies (typically 100 eV) on the substrate’s surface. The as-landed particles start to form chemical bonds with each other at the molecular level, forming a continuous thin film with the same composition as the target material [55,56]. Further laser pulses ablate more material, and the growing layer’s thickness will increase from monolayers up to microns. The layers grow through diffusion and particle aggregation, which improves the layer-by-layer growth [55,57]. According to the literature, the laser type most commonly used to fabricate CZTS absorber layers are excimer lasers and Nd:YAG lasers. 

Despite the unique beneficial features offered by the PLD in terms of the high-quality growth layer, the literature does not abound in reported results about the notable performance of CZTS solar cell devices processed by PLD. The PLD equipment is expensive and this strongly restricts the marketability of PLD-manufactured photovoltaic devices.

The first CZTS thin-film solar cell prepared by PLD was reported in 2007 by Moriya et al. The authors reported a V_OC_ of 546 mV and a conversion efficiency of 1.74% for their solar cell based on an absorber CZTS synthetized by PLD at 500 °C in N_2_. Moholkar et al. [58] synthetized thin films of CZTS by PLD that were used as an absorber layer in a solar cell configuration. The fabricated solar cell exhibited an open-circuit voltage of 585 mV and a conversion efficiency of 2.02%. Using the PLD technique, in 2012, the same group of authors reported the impact of the chemical composition ratio of the target Cu/(Zn + Sn) on the performance of the final solar cell device. They found that by increasing the Cu/(Zn + Sn) ratio from 0.8 to 1.2 while keeping the Zn/Sn constant, the direct bandgap energy of the CZTS absorber decreased from 1.72 eV to 1.53 eV. The best performances reported by the authors were a conversion efficiency of 4.13% and *V_OC_* = 700 mV on a glass/Mo/CZTS/CdS/ZnO:Al/Al solar cell fabricated by using Cu/(Zn + Sn) = 1.1 [59,60,61].

### 3.2. Chemical Methods

The chemical methods of depositing thin films are the most economically attractive and versatile techniques for growing sulfide and oxide layers. The most notable features of this method are its ease of handling, the simplicity of the process, the compact low-cost equipment, and low wastage of the precursor materials. However, some critical issues have to be addressed when the chemical deposition methods of absorber CZTS layers are used, such as the need for multiple uniform coatings, uniformity, and cracks. 

The current record for the efficiency of CZTS solar cells (12.6%) was found for a CZTS absorber layer processed by the chemical solution approach using hydrazine as a solvent [3]. As this solvent is hazardous, reactive, and unstable, the hydrazine-based chemical method is not attractive to the industry for large-scale CZTS solar cell fabrication.

#### 3.2.1. Sol-Gel Method

Sol-gel, based on the spin coating technique, is a simple and cost-efficient approach to the preparation of various thin oxide films. This method is very attractive for mass-production because of its easy handling and its suitability for large-area deposition, and because no vacuum system is required, allowing thermal annealing in air at temperatures as low as room temperature. Another remarkable feature of the sol-gel synthesis method is that this technology is suitable for the synthesis of high-yield, high-quality solar cells made from thin CZTS thin films through inexpensive green processes, making it very attractive for large-scale industrialization [62,63]. However, some issues are related to this method; for example, it is a time-consuming process and controlling the porosity of the growing layer is quite difficult.

The sol-gel method consists of three main steps, i.e., (i) preparation of the precursor solution and spin-coating the precursor solution onto the substrate, (ii) the low-temperature thermal treatment, and (iii) the final high temperature annealing. Figure 8 shows a typical set-up of synthesizing the CZTS precursor solution.

The CZTS precursor solution is synthetized by dissolving the elements raw precursors in 2-methoxyethanol (C_3_H_8_O_2_), i.e., a salt of copper (for Cu), a salt of zinc (for Zn), a salt of tin (for Sn), and thiourea (NH_2_)_2_CS (for S). The 2-methoxyethanol solvent removes the necessity to perform prolonged low-temperature processing of the deposited solution on the substrate after spin-coating (110 °C ÷ 120 °C temperature annealing for 80 ÷ 120 min), which not only significantly increases the fabrication time of the CZTS film at the optimal thickness (around 1 µm) but also favors the formation of undesirable oxide compounds in CZTS films, and thus results in the low reproducibility of the electrical and optical parameters [64,65,66,67]. Dimethyl sulfoxide (DMSO) has been successfully used for the synthesis of the precursor solution and spin-coating of CZTS films. The most prominent features of the DMSO solvent relate to its green technology approach, its non-toxic character, the high yield, and the low cost [67].

Tanaka et al. synthesized CZTS films by spin-coating a sol-gel precursor solution based on copper (II) acetate monohydrate, zinc (II) acetate dehydrate, and tin (II) chloride dehydrate dissolved in a mixed solution of a 2-methoxyethanol (2-metho) solvent and a monoethanolamine (MEA) stabilizer [68]. Thin CZTS films were grown by annealing the precursor films in an atmosphere containing H_2_S at 500 °C. The fabricated CZTS solar cells had an open-circuit voltage of 390 mV and an efficiency of 1.01%. The same group of authors investigated the impact of the precursor solution’s chemical composition on the morphology and the photovoltaic performance of the thin CZTS films fabricated by the sol-gel spin-coating technique. The authors varied the Cu/(Zn + Sn) ratio in precursor solution of the sol-gel while keeping the Zn/Sn ratio constant at 1.15 [69]. According to the results reported by Tanaka, large grains characterized the films made from the copper-poor precursor solution (Cu/(Zn + Sn) < 0.8, molar ratio). The best solar cell had a *V_OC_* of 575 mV and an efficiency of 2.03%. Agawane et al. reported achieving a 0.77% conversion efficiency with a solar cell fabricated from a CZTS absorber layer synthetized by the sol-gel technique annealed under S powder [70]. Later on, the same group of authors reported an improvement in the processing conditions and achieved a conversion efficiency of 3.01% with a solar cell based on a CZTS absorber layer prepared by a non-toxic, simple, and economical sol-gel and spin-coating technique annealed at 550 °C under a H_2_S gas atmosphere [62]. Using non-aqueous thiourea–metal–oxygen sol-gel processing, Su et al. synthetized high-quality phase-controlled CZTS layers with a homogeneous elemental distribution and a low impurity content by introducing Na as an extrinsic dopant [63]. Through use of this modified sol-gel method, the fabricated Ni:Al/ZAO/i-ZnO/CdS/1%Na:CZTS/Mo/glass solar cells achieved a power conversion efficiency of 5.10%, while the device without Na had an efficiency of only 4.10%. The authors reported that a further improvement in the conversion efficiency of up to 5.10% was obtained by HCl chemical etching of the CZTS absorber layer. Liu et al. used the sol-gel and selenization processes and fabricated high-quality single-phase kesterite CZTSSe with a total area efficiency of 8.5% without an antireflection coating [71]. The sol-gel method was also used in the study performed by Sun et al. to fabricate solar cells based on CZTS absorbers [72]. The authors tuned the Cu/Sn with the purpose of improving the films’ uniformity, reducing the formation of voids, and enhancing the power conversion efficiency. The best cell achieved an efficiency of 8.8% with an open-circuit voltage of 746 mV. Todorov et al. processed CZTS films by spin-coating deposition of a precursor solution of a slurry of hydrazine and particles of the Cu, Zn, Sn, and S elements. The spin-coated CZTS precursor thin films were annealed at 540 °C in an atmosphere containing sulfoselenide, and the best measured efficiency was 9.66% [73]. In 2013, Wang et al. all reported the results of their own approach involving the liquid deposition of thin CZTS films based on a pure solution of hydrazine [3]. The reported results were independently certified and the conversion efficiency of 12% they achieved represents the world-record power conversion efficiency for CZTSSe thin films. The new device’s open-circuit voltage was 513.4 mV.

#### 3.2.2. The Solvothermal Method

The solvothermal method is a chemical process in which the chemical reaction between the involved precursors take place in a solvent sealed in a vessel heated above the solvent’s boiling temperatures (usually less than 250 °C). The sealed reaction vessel, also called an autoclave, acts as a closed system in which the elevated temperature and pressure promote the reactivity of the reactants, leading to the formation of a supercritical fluid in which the gas and liquid phases coexist, favoring the crystallization of the dissolved raw precursors. The solvothermal reaction products are homogenous high-purity nanosized particles. The products’ shape and size are controlled through the reaction temperature, solvent, and pH. The solvothermal method does not require post-synthesis annealing processes of the resulting products. The resulting nanoparticles are further used to grow thin films. In this method, the working substrate is immersed in a mixed solution of the solvent and the solvothermal products dissolved in it. After dipping, the covered substrate is dried in order to remove the solvent. The thin films grown by the solvothermal method have good uniformity, high crystallinity, and a stable structure. The main factors affecting the thin films’ growth are: (i) the nature of solvent, (ii) the temperature, (iii) the time, (iv) the concentration of the precursor material, (v) the concentration of OH− ions in the solution, and (vi) the pH. One of the most notable features of the thin films fabricated by the solvothermal method is the low-temperature growing process that reduces the possibility of chemical reactions between the substrate and the reactants. Other benefits offered by the solvothermal method of synthesizing CZTS absorbers include its ease of use, and its accurately and reliability in synthesizing high-purity and highly crystalline materials. This method is considered to be a green technology that is suitable for mass production of good-quality crystals while maintaining the control over their composition and morphology [74].

Several research groups have reported results for CZTS-based solar cells fabricated by the solvothermal method [75,76,77,78]. To obtain CZTS absorber layers, firstly the solvothermally synthesized CZTS nanoparticles are dispersed in solvents to create a paste or ink. The as-obtained slurry is deposited on the working substrate by printing, spraying, or dip-coating, followed by a high-temperature thermal annealing treatment to ensure the formation of the crystalline phase [79,80]. 

Wei et al. used the solvothermal method to grow Cu_2_ZnSnS_4_ (CZTS) layers directly on transparent conductive fluorine-doped tin oxide (FTO) substrates using hexadecyl trimethyl ammonium bromide (CTAB) as the surfactant [81]. The fabricated glass/FTO/CZTS/CdS/i-ZnO/AleZnO/Ag solar cells were characterized by an optical bandgap of 1.52 eV and a poor power conversion efficiency of only 0.16%. Using the solvothermal method, Al-Hadeethi et al. studied the impact of the concentration of triethanolamine (TEA) on the structure, phase formation, morphology, and composition of the resulting CZTS nanoparticles [78]. The authors fabricated a SLG/Mo/CZTS/CdS/i-ZnO/Al:ZnO/Al structure using CZTS particles synthesized with a TEA concentration of 5 mg/mL, and the best demonstrated solar conversion efficiency was reported to be 4.33%. In a recent study published by Kannan et al., the authors used ethylene glycol (EG) as a solvent and investigated the impact of the Cu/(Zn + Sn) ratio on the properties of the CZTS nanoparticles synthesized by the solvothermal method [82]. The synthetized nanocrystals were used to fabricate a CZTS/ZnO photovoltaic structure from a scalpel-cut absorber and a drop-cast ZnO buffer layer. The authors reported that the devices based on absorber layers that were slightly poor in Cu (0.9) provided the best open-circuit voltage (510 mV), short circuit current (11.49 mA/cm^2^), filing factor (52.97%), and power conversion efficiency (3.2%).

## 4. Concluding Remarks about the Impact of Processing on the V_OC_ Deficit in CZTS Solar Cells

In this review, the current issues related to the low open-circuit voltage deficit dominating the performance of CZTS solar cells have been discussed. Although significant progress in engineering the CZTS absorber layer has been achieved, up to now, the 12% efficiency milestone for kesterite solar cells has not yet been surpassed.

As generally accepted, the primary factor diminishing the photovoltaic performance of CZTS-based solar cells is the high open-circuit deficit. This review has presented and discussed the main approaches adopted to lower this detrimental parameter. Based on the relationship between the V_OC_ and the bandgap energy, several approaches of lowering the ΔV_OC_ through tuning the Eg have been reviewed, including the partial substitution of S by Se and the Cu/(Zn + Sn) ratio. According to the reviewed reports, it is obvious that while the Eg increases monotonically with the Se content, the V_OC_ increases up to a certain level of Se doping, after which a drop in *V_OC_* will occur. As the chemical composition has to be controlled in the very narrow stability window of the pure kesterite crystal phase, the authors suggest that the most beneficial approach should be related to implementing artificially engineered CZTS absorber materials based on continuously or step-graded chemical gradients, as obtained by tuning the composition of CZTS from Cu-poor/Zn-rich to Cu-rich/Zn-poor in the same layer.

Finally, the most attractive mass production fabrication methods of CZTS absorbers were reviewed with the aim of indicating the current performance, the advantages, and the weaknesses of each of these methods. 

## Figures and Tables

**Figure 1 materials-15-08427-f001:**
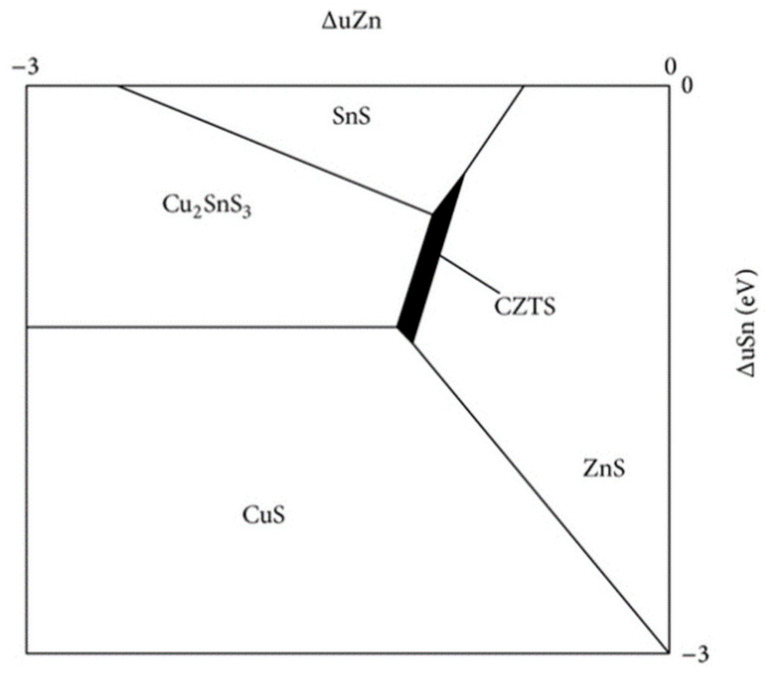
Chemical potential phase diagram of CZTS under Cu-rich conditions [26].

**Figure 2 materials-15-08427-f002:**
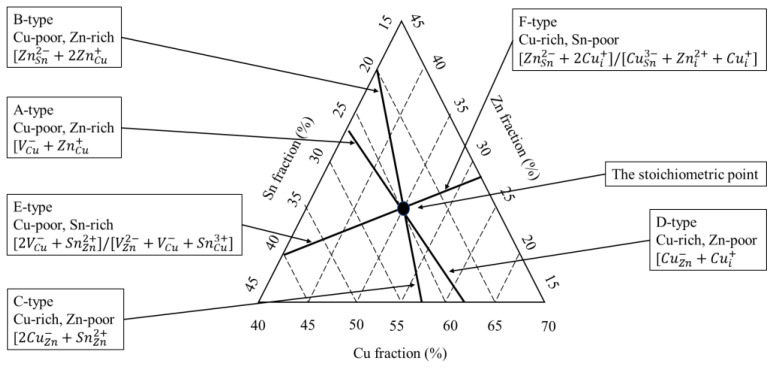
Compositions of the defined types of defect complexes in CZTS (adapted from [33]). The stoichiometric composition point is given by the intersection of all these lines.

**Figure 3 materials-15-08427-f003:**
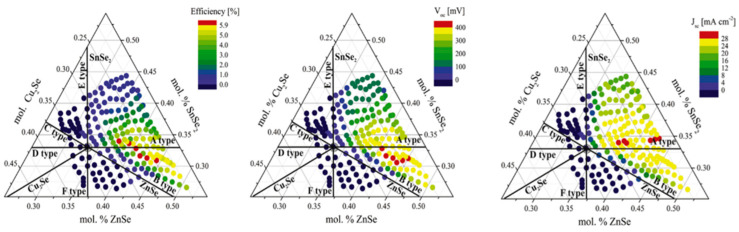
Combinatorial experiment showing the relationship of conversion efficiency and V_OC_ with the composition of CZTS. Reprinted with permission from [10]. Copyright Elsevier.

**Figure 4 materials-15-08427-f004:**
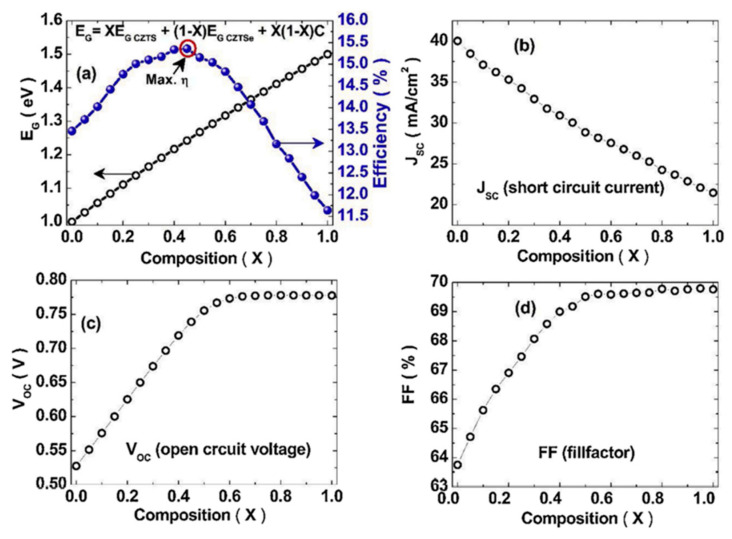
Calculated variation in (**a**) *E_g_* and efficiency, (**b**) short circuit current *J_SC_*, (**c**) open-circuit *V_OC_*, and (**d**) the filling factor (FF) upon the Se content in a Cu_2_ZnSn(S_1−X_Se_X_)_4_ absorber as calculated by Kumar et al. Reprinted with permission from [8]. Copyright Elsevier.

**Figure 5 materials-15-08427-f005:**
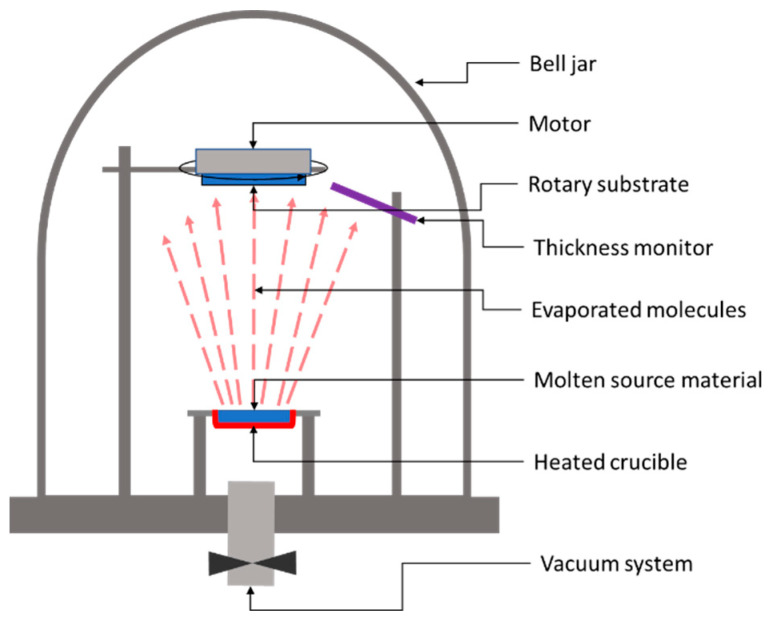
Schematic representation of a classic thermal evaporation system.

**Figure 6 materials-15-08427-f006:**
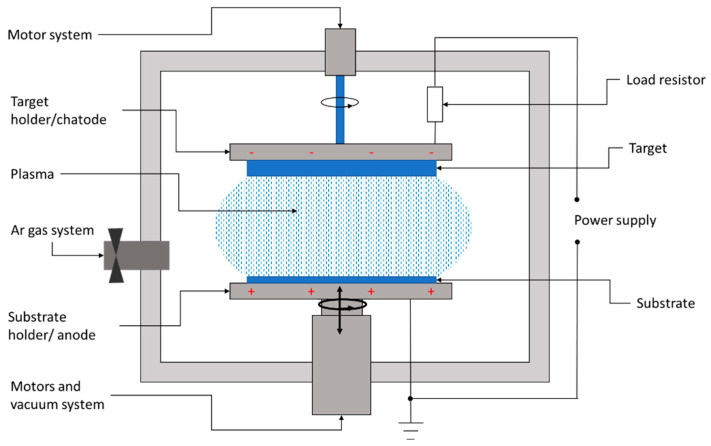
Layout of a DC sputtering system for the deposition of thin films.

**Figure 7 materials-15-08427-f007:**
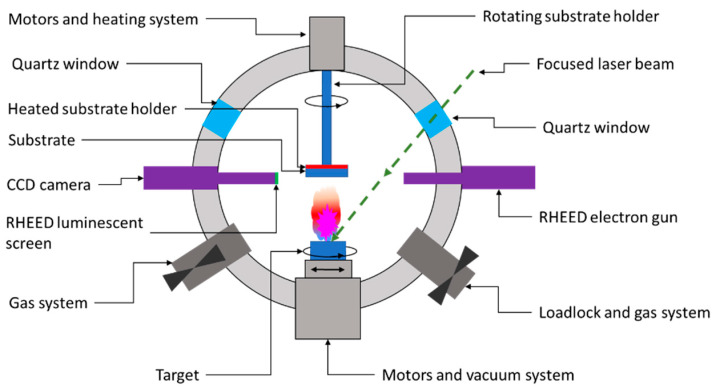
Schematic representation of a PLD deposition chamber.

**Figure 8 materials-15-08427-f008:**
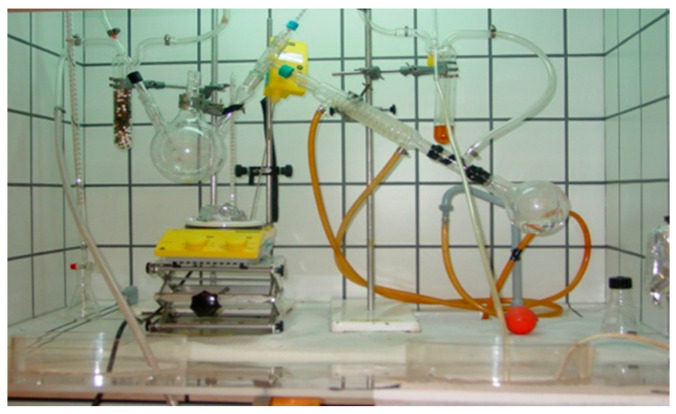
Installation used for the synthesis of the CZTS precursor solution at the authors’ facilities.

## Data Availability

Not applicable.

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
