# Peer review of "Current Status of the Open-Circuit Voltage of Kesterite CZTS Absorber Layers for Photovoltaic Applications—Part I, a Review"

_materials, 2022, doi:10.3390/ma15238427_

Round 1

Reviewer 1 Report

Current status of open circuit voltage of Kesterite CZTS ab- 2sorber layer for photovoltaic applications – Part I, a review

In my opinion, this work is not enough for a review.

The physical origin of low Voc is not addressed. Just commented but no discussions, and there are no new proposals or some light.

There are no discussions of each layer and the neighbor effect between them.

Just two or three lines of paper that report physical and chemical techniques to deposit CZTS but not deep scientific analysis. Furthermore, no conductivity analysis of the layers including hall effect discussions. Depletion region analysis and no back surface field review were reported. So, see this reference which could be useful to enhance the work:

Martín G. Reyes-Banda, et al, Effect of Se diffusion and the role of a thin CdS buffer layer in the performance of a CdSe/CdTe solar cell, Superlattices and Microstructures, Volume 133, 2019, 106219, ISSN 0749-6036, https://doi.org/10.1016/j.spmi.2019.106219.

Guang-Xing Liang, Yan-Di Luo, Ju-Guang Hu, Xing-Ye Chen, Yang Zeng, Zheng-Hua Su, Jing-Ting Luo, Ping Fan, Influence of annealed ITO on PLD CZTS thin film solar cell, Surface and Coatings Technology, Volume 358, 2019, Pages 762-764, ISSN 0257-8972, https://doi.org/10.1016/j.surfcoat.2018.11.079.

Gutierrez Z-B, F. et al. Development of a CdCl2 thermal treatment process for improving CdS/CdTe ultrathin solar cells. J Mater Sci: Mater Electron 30, 16932–16938 (2019). https://doi.org/10.1007/s10854-019-01694-2

What happened with the absorption coefficient as a function deposition technique? That’s important for the thickness of the absorber layer?

What’s the effect of grain size and how it impacts the Vco?

The effect of Jsc is not discussed.

Reverse saturation current or leakage current is not discussed. It is key for Voc.

The conclusion is so poor.

Too much discussion is required for a review. A review is not just to mention a few of the published papers.

Author Response

Manuscript ID: materials-2048236

Answers to Editor and Reviewers

Authors are grateful to Editor and Reviewers for their comprehensive analyses of our previous text and the critical but constructive comments and recommendations, which we did the best to consider in the revision of the manuscript.

Namely, we answered point by point to your observation on the text and all changes were indicated by track-changes in the revised manuscript.

All revisions made to the manuscript have be marked up using the “Track Changes” function of MS Word.

Reviwer #3

Current status of open circuit voltage of Kesterite CZTS ab- 2sorber layer for photovoltaic applications – Part I, a review

In my opinion, this work is not enough for a review.

The physical origin of low Voc is not addressed. Just commented but no discussions, and there are no new proposals or some light.

  1. There are no discussions of each layer and the neighbor effect between them.

Answer: As mentioned in the text, the paper focus is on the CZTS bulk properties. The others layers and the neighbor effect between them is not in the scope of the first part of this review. In this review, we have not discussed about the cell architecture, and the strength and witnesses of each of them.

  1. Just two or three lines of paper that report physical and chemical techniques to deposit CZTS but not deep scientific analysis. Furthermore, no conductivity analysis of the layers including hall effect discussions. Depletion region analysis and no back surface field review were reported. So, see this reference which could be useful to enhance the work:
  2. Martín G. Reyes-Banda, et al, Effect of Se diffusion and the role of a thin CdS buffer layer in the performance of a CdSe/CdTe solar cell, Superlattices and Microstructures, Volume 133, 2019, 106219, ISSN 0749-6036, https://doi.org/10.1016/j.spmi.2019.106219.
  3. Guang-Xing Liang, Yan-Di Luo, Ju-Guang Hu, Xing-Ye Chen, Yang Zeng, Zheng-Hua Su, Jing-Ting Luo, Ping Fan, Influence of annealed ITO on PLD CZTS thin film solar cell, Surface and Coatings Technology, Volume 358, 2019, Pages 762-764, ISSN 0257-8972, https://doi.org/10.1016/j.surfcoat.2018.11.079.
  • Gutierrez Z-B, F. et al.Development of a CdCl2thermal treatment process for improving CdS/CdTe ultrathin solar cells. J Mater Sci: Mater Electron 30, 16932–16938 (2019). https://doi.org/10.1007/s10854-019-01694-2

Answer: We found the suggested i) and iii) references as referring to CdS/CdTe solar cells. Moreover, it is worth noting that this type of photovoltaics, as well CGIS, are involving CRM. The scope of the author paper is CZTS. The suggested reference ii), even if is referring to CZTS, is based on discussed about some processing conditions on the performances of CZTS solar cells. Even if these paper reports about an efficiency improvement up to only 1.92% - just below of the first maturity check point of 2% - the author has included this reference in the Processing part of its revised version.  The author thanks to R#3 for the suggested references i), and iii).

  1. What happened with the absorption coefficient as a function deposition technique? That’s important for the thickness of the absorber layer?

Answer: At the paper’ line 216, the optical gap is introduced. As concerning to the R#3 asked question about the Eg-n relationship, due to the complexity of this subject, e.g. investigation techniques (UV-VIS-NIR, spectroscopic ellipsometry, photoluminescence, etc), the n’ dependence of the photon energy, recombination processes, mathematical models to be involved – for instance even if the Sellmeier’ equation could be applied, the n dispersion is not related to the Kramers-Kroning relations; or applying the transfer matrix formalism to extract the complex refractive index through the Maxwell equations under the boundary conditions; or simply involving spectroscopic ellipsometry – this matter will be discussed elsewhere toghether with the authors results on continuously/step-graded CZTS absorbers.

  1. What’s the effect of grain size and how it impacts the Vco?

Answer: This is discussed at the line 40, 62 and 88 of the revised version.

  1. The effect of Jsc is not discussed.

     Answer: The authors has focused the review attention on the possible origins of the characteristic large loss in the open circuit voltage as the main factor of diminishing the CZTS performances. However, we added to the revised version supplementary remarks about the JSC and FF through the new version of Figure 3.

  1. Reverse saturation current or leakage current is not discussed. It is key for Voc.

     Answer: The scope of this Part I of the review is to give a knowledge update about the attempts for reducing the open circuit voltage deficit of the CZTS photovoltaics in respect with the absorber bulk material properties.

  1. The conclusion is so poor.
  2. Answer: The author has improved the conclusion part.

  1. Too much discussion is required for a review. A review is not just to mention a few of the published papers.

     Answer: The author is fully agreeing the R#3, and as planned, this part I will be continued by part II and III.

Author Response

Manuscript ID: materials-2048236

Answers to Editor and Reviewers

Authors are grateful to Editor and Reviewers for their comprehensive analyses of our previous text and the critical but constructive comments and recommendations, which we did the best to consider in the revision of the manuscript.

Namely, we answered point by point to your observation on the text and all changes were indicated by track-changes in the revised manuscript.

All revisions made to the manuscript have be marked up using the “Track Changes” function of MS Word.

Reviewer #2:

In this review, the authors have done remarkable job reviewing reduce the open circuit voltage of the CZTS kesterite based-solar cells performances. This review will also present the strengths and weakness of the most commercially-at-atractive synthesis methods for CZTS kesterite thin films synthesis for photovoltaic application. The research topic and overall quality of the paper fall within the scope of Materials. Before the paper could proceed further for publication, the authors are advised to address the following issues

Q1: Many paragraphs are too short to read in the manuscript. It would be better if the author could combine several short paragraphs into a long paragraph.

Answer: As Reviewer requested, the author has combined several short paragraphs into longer paragraph.

Q2. For solar cells, short-circuit current and filling factor has always been the focus of attention, so it is recommended to increase this discussion to the manuscript.

Answer: The authors has focused the review attention on the possible origins of the characteristic large loss in the open circuit voltage as the main factor of diminishing the CZTS performances. However, we added to the revised version supplementary remarks about the JSC and FF through the new version of Figure 3.

Q3. Because solvothermal method has some unique properties, such as high pressure (J. Phys. Chem. Lett. 2020, 11, 98620), more discussion on solvothermal method is expected to be given in the manuscript.

Answer: The author has not able to identify the suggested reference. However, the author is developing a research on graded CZTS solar cells by involving the solvothermal and sol-gel techniques as for absorber fabrication methods. Accordingly, deep description of both methods are planned to be released into the second part of here discussed paper.

Q4. There are many awkward usages and errors in the manuscript, for example, subheading 4 is wrongly written as subheading 3 (page 12), resulting in two subheadings 3 in the manuscript. In addition, the logic of the abstract is very confused and lacks a good summary.

Answer: The author has corrected the awkward usages and errors in the manuscript.

Q5. In this review, there are few charts, especially in the subheading 3, there is almost no one. It is suggested that the author elaborate some published works in the form of combination diagram.

Answer: The author has added Figure 5, 6, 7 and 8 to the subheading 3.

Q6. The quality of Figure 1-2 is poor and unclear. The authors should improve them before the publication of this work.

Answer: The author has improved the Figure 1-2 resolution.

Reviewer 3 Report

Dear Editor,

The paper Current status of open circuit voltage of Kesterite CZTS absorber layer for photovoltaic applications – Part I, a review. There are a few misses that require some modifications in the paper. After the following corrections paper may be acceptable for publication:

1. The Abstract should be revised carefully and highlight the new results achieved in the literature.

2.The English and grammatical mistakes should be revised carefully and the monoculture.

3. the Introduction part needs to be improved with specifications Interestingly, I suggested that authors start talking about the different technologies and add part abourt perovskite and tandem perovskite and why they chose to investigate  CZTS, and using updated references

Suggested references,

https://doi.org/10.4995/Thesis/10251/160621, https://doi.org/10.1016/j.solener.2019.12.016

4. I recommend in this paragraph can be improved by authors and there are some new technology tandem CZTS/perovskite :[ The worldwide champion efficiency of 12.6% is reported for the CZTS absorber layer  processed by the chemical solution approach using hydrazine as a solvent [2]. Due to the 266 fact that hydrazine is a hazardous, reactive, explosive and unstable solvent, the large scale 267 CZTS solar cells fabrication is critical limited.] suggested references: https://doi.org/10.1016/j.ijleo.2022.168854

5. I suggest that the authors could improve Processing technologies for kesterite CZTS absorber thin films by adding some schemas.

 6. conclusion should be improved .

Author Response

Manuscript ID: materials-2048236

Answers to Editor and Reviewers

Authors are grateful to Editor and Reviewers for their comprehensive analyses of our previous text and the critical but constructive comments and recommendations, which we did the best to consider in the revision of the manuscript.

Namely, we answered point by point to your observation on the text and all changes were indicated by track-changes in the revised manuscript.

All revisions made to the manuscript have be marked up using the “Track Changes” function of MS Word.

Reviewer #1

Dear Editor,

The paper Current status of open circuit voltage of Kesterite CZTS absorber layer for photovoltaic applications – Part I, a review. There are a few misses that require some modifications in the paper. After the following corrections paper may be acceptable for publication:

  1. The Abstract should be revised carefully and highlight the new results achieved in the literature.

Answer: The author has improved the Abstract part by highlighting the review content.

  1. The English and grammatical mistakes should be revised carefully and the monoculture.

Answer: We carefully checked all text, improved the translation and repaired the typo errors.

  1. The Introduction part needs to be improved with specifications Interestingly, I suggested that authors start talking about the different technologies and add part about perovskite and tandem perovskite and why they chose to investigate CZTS, and using updated references.

Suggested references,

https://doi.org/10.4995/Thesis/10251/160621, https://doi.org/10.1016/j.solener.2019.12.016

Answer: The Introduction part was improved and the 2 suggested references were included in the text.

  1. I recommend in this paragraph can be improved by authors and there are some new technology tandem CZTS/perovskite:

[ The worldwide champion efficiency of 12.6% is reported for the CZTS absorber layer processed by the chemical solution approach using hydrazine as a solvent [2]. Due to the 266 fact that hydrazine is a hazardous, reactive, explosive and unstable solvent, the large scale 267 CZTS solar cells fabrication is critical limited.] suggested references: https://doi.org/10.1016/j.ijleo.2022.168854

Answer: The paragraph was improved, and the suggested reference was included.

  1. I suggest that the authors could improve Processing technologies for kesterite CZTS absorber thin films by adding some schemas.

Answer: The author has improved the Processing part by adding 3 new figures.

  1. Conclusion should be improved.

Answer: The author has improved the conclusion part.                    

Round 2

Reviewer 1 Report

I do not recommend this work for publication.

The paper is not adequate for a review because the Voc is related to more physical aspects such as the absorption coefficient, Jsc, and FF. The recombination in each layer, the grain size, and the Ohmic contacts, all of them affect the Voc. So, the bulk paradigm is not enough to discuss the Voc.

Furthermore, the chemical techniques are not well discussed, SILAR method is not included in the review.

Finally, chemical analysis such as XPS and SIMS characterization of the CZTS solar cell was missing in the review.
